# Hybrid Manufacturing Processes Used in the Production of Complex Parts: A Comprehensive Review

Naiara P. V. Sebbe [1], Filipe Fernandes [1,2,*], Vitor F. C. Sousa [1] and Francisco J. G. Silva [1,3]

1    ISEP—School of Engineering, Polytechnic of Porto, Rua Dr. António Bernardino de Almeida 431, 4200-072 Porto, Portugal
2    CEMMPRE—Centre for Mechanical Engineering Materials and Processes, Department of Mechanical Engineering, University of Coimbra, Rua Luís Reis Santos, 3030-788 Coimbra, Portugal
3    INEGI—Driving Science and Innovation, Rua Dr. Roberto Frias 400, 4200-465 Porto, Portugal
*    Correspondence: fid@isep.ipp.pt

**Abstract:** Additive manufacturing is defined as a process based on the superposition of layers of materials in order to obtain 3D parts; however, the process does not allow achieve the adequate and necessary surface finishing. In addition, with the development of new materials with superior properties, some of them acquire high hardness and strength, consequently decreasing their ability to be machined. To overcome this shortcoming, a new technology assembling additive and subtractive processes, was developed and implemented. In this process, the additive methods are integrated into a single machine with subtractive processes, often called hybrid manufacturing. The additive manufacturing process is used to produce the part with high efficiency and flexibility, whilst machining is then triggered to give a good surface finishing and dimensional accuracy. With this, and without the need to transport the part from one machine to another, the manufacturing time of the part is reduced, as well as the production costs, since the waste of material is minimized, with the additive–subtractive integration. This work aimed to carry out an extensive literature review regarding additive manufacturing methods, such as binder blasting, directed energy deposition, material extrusion, material jetting, powder bed fusion, sheet laminating and vat polymerization, as well as machining processes, studying the additive-subtractive integration, in order to analyze recent developments in this area, the techniques used, and the results obtained. To perform this review, ScienceDirect, Web of Knowledge and Google Scholar were used as the main source of information because they are powerful search engines in science information. Specialized books have been also used, as well as several websites. The main keywords used in searching information were: "CNC machining", "hybrid machining", "hybrid manufacturing", "additive manufacturing", "high-speed machining" and "post-processing". The conjunction of these keywords was crucial to filter the huge information currently available about additive manufacturing. The search was mainly focused on publications of the current century. The work intends to provide structured information on the research carried out about each one of the two considered processes (additive manufacturing and machining), and on how these developments can be taken into consideration in studies about hybrid machining, helping researchers to increase their knowledge in this field in a faster way. An outlook about the integration of these processes is also performed. Additionally, a SWOT analysis is also provided for additive manufacturing, machining and hybrid manufacturing processes, observing the aspects inherent to these technologies.

**Keywords:** hybrid manufacturing; CNC machining; additive manufacturing; 5-axis machining; manufacturing processes

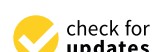



## 1. Introduction

Nowadays, with the advances in materials for several industrial sectors, such as the aerospace, biomedical and automotive sectors, new methods/processes are needed to fulfil

the production demand and required quality [1]. Standard methods of manufacturing have thus started to be put aside, whilst more advanced methods of manufacturing raised, so that customer needs are met without affecting the profitability of the companies [2].

It is in this way that the concept of hybrid manufacturing arises. The idea behind hybrid manufacturing is joining different processes on the same setup in order to achieve the effect known as "1 + 1 = 3" [3,4]. The term "hybrid manufacturing" is directly linked to the integration of different processes, and its development is related to the requirements and complexity of new parts [5].

Thus, the objective of hybrid manufacturing is the joining of two or more distinct processes in a single piece of equipment, observing the unique advantages of each one, while minimizing the limitations of the process [6]. It is observed that there are several types of hybrid manufacturing, although the most common is the type that combines laser additive manufacturing and 5-axis machining processes [7].

Additive manufacturing allows the production of 3D geometric parts through layer overlay [8,9], being a suitable method for efficient production of parts; however, the cost is high and requires a high financial capital, which makes the technique not yet widely used [10]. 5-axis machining is a process more and more used in the industry [11], allowing a single configuration to machine five sides of the part [12] and providing high accuracy and surface quality [13].

The present work aims to carry out a broad bibliographic review about hybrid manufacturing regarding additive and subtractive processes in a single equipment. Additive manufacturing and its manufacturing processes, such as binder blasting, directed energy deposition, material extrusion, material jetting, powder bed fusion, sheet laminating and vat polymerization, and their respective subclassifications were thoroughly and properly analyzed and described. An approach to machining is also taken, emphasizing 5-axis machining. From this, hybrid manufacturing is discussed through a broad review of the work carried out and the integration between additive and subtractive processes, as well as the challenges faced and future opportunities.

## 2. Additive Manufacturing

Additive manufacturing is the process of manufacturing parts layer by layer, enabling the production of parts with more robust and complex geometries [14]. This technology is of great relevance and represents a real challenge for today's industries, given its flexibility and ability to provide differentiated products and parts [15–18]. Additive manufacturing is popularly known as "3D printing", and it is a process which takes place under digital control. In this process, the raw material is placed in the equipment in the form of wire or powder, and the part produced according to the geometry contained in a CAD project (computer-aided design) [19]. Figure 1 exposes the basic principle of additive manufacturing of a part grown layer by layer [20].

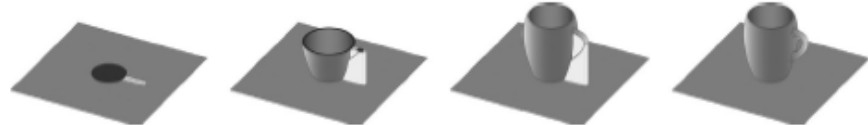

**Figure 1.** Basic principle of additive manufacturing. Reproduced from [20], 2012, Elsevier.

The classification of additive manufacturing processes, in the past, was based on the criterion of separating materials into a liquid basis, solid basis or powder basis [21]. However, in 2010, the American Society for Testing and Materials (ASTM) published the standard "ASTM F42—Additive Manufacturing", which considers additive manufacturing in seven categories [22]:

(i)　　Binder blasting;
(ii)　 Directed energy deposition—DED (comprised of processes such as direct deposition of metal), such as wire arc additive manufacturing (WAAM);

(iii)   Material extrusion (which includes fuse deposition modeling (FDM));
(iv)   Material jetting;
(v)    Powder bed fusion (which includes processes such as direct metal laser sintering (DMLS) and selective laser sintering (SLS));
(vi)   Sheet laminating (ultrasonic additive manufacturing (UAM) and laminated object manufacturing (LOM));
(vii)  Vat polymerization.

In binder blasting, an ink jet is selectively deposited onto bond powder materials, typically plaster or starch, and creates three-dimensional objects, consisting of placing thin layers of this powder, with the print head ejecting and depositing drops of binder [23]. In the study carried out by Chen et al. [24], alumina powder was used as a material processed in different ways to obtain granules with different properties, and aiming at the final quality of the part in terms of compressive strength and density. With this, it was verified that the alumina samples were produced properly and with density and compressive strength within the expected ranges.

The directed energy deposition (DED) process uses thermal energy to build the 3D product, layer by layer, with very good properties. The volumetric density of the product can be practically 100%, and its use with hybrid systems is very frequent, given the ability to deposit heterogeneous materials on the substrate with adequate characteristics [25–27]. Figure 2 illustrates the DED process.

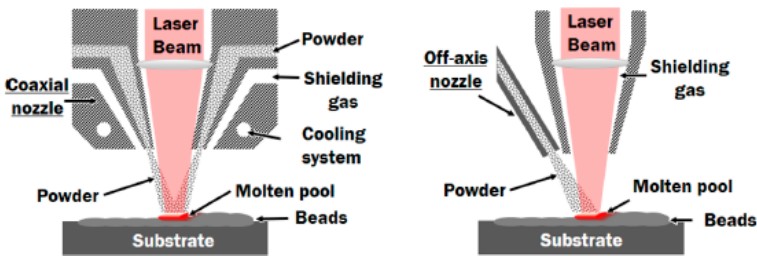

**Figure 2.** DED process illustration. Reproduced from [27]. Creative Commons CC BY license: https://creativecommons.org/licenses/by/4.0/; accessed on 21 October 2022.

In the work by Chen et al. [28], it was observed that changes in temperature directly influence the melting and cooling of the material, and consequently also influence the final microstructure and the resulting hardness. Thus, it was possible to predict the hardness distribution of a part by monitoring the temperature changes that occurred. For this purpose, some key temperature features (KTFs) were defined, according to the temperature field present in the DED process, and the resulting hardness predictions were made, showing that predictions were consistent with the real hardness trends.

Tekumalla et al. [29] studied high-vanadium high-speed steels (HVHSSs), which are considered difficult to machine materials due to their high hardness. Thus, the DED method was used, with two alloy compositions: Fe-10V-4.5Cr-2.5C and Fe-15V-13Cr-4.5C, which are highly wear-resistant alloys due to their vanadium and carbon content, and after processing, high hardness of the products was confirmed through a comparison performed regarding these materials after conventional heat treatment. In the work by Radhakrishnan et al. [30], the laser-directed energy deposition method was used to deposit titanium/titanium carbide (Ti/TiC) with 20%, 40% and 60% TiC. The formation of the non-stoichiometric compound TiC0.55 was observed, and hardness increased with TiC content increasing.

Still regarding the DED process, wire feeding processes are divided into three types: (i) wire and arc additive manufacturing (WAAM) processes, which combine an arc, working as a thermal energy source, and a wire, which is the raw material, providing energy efficiency less than 90%; (ii) additive manufacturing of wire and laser (WLAM), which in turn uses the concept of laser cladding and welding in order to produce metal parts that

do not have porosity; and finally, (iii) additive manufacturing of wire and electron beam (WEAM), which, as the name implies, uses a beam of electrons as a source of energy [27].

Grossi et al. [31] analyzed the dynamic behavior of a NACA 9403 airfoil produced using the WAAM technique, exploring by finite element analysis (FE) the subsequent machining process in order to predict the dynamics of the part during that machining process. Thus, the system was able to simulate the material removal process and the dynamic behavior of the part previously generated by WAAM, updating its variable geometry.

On the other hand, using another technology known as PAW (plasma arc welding), and in order to analyze the deposition process based on process characteristics and mechanical and microstructural properties, Artaza et al. [32] produced two Ti6Al4V walls with a high deposition rate (2 kg/h) and in an inert argon atmosphere, with no significant differences in microstructural or mechanical properties after thermal heat treatments. Still in relation to PAW technology and Ti6Al4V alloy, Veiga et al. [33] analyzed the production, as well as the quality, of the alloy produced using this technology with subsequent milling, and observed that there were no large deviations in the mechanical properties of the samples in different positions and orientations, and the up-milling showed torque values that were slightly larger. However, the quality of the final surface was superior.

The principle of material extrusion is a technique widely used popularly in domestic environments. This process uses a wire which is extruded and gradually forms the product. A layer of material is deposited horizontally through a nozzle, and then with increments in the vertical direction, until the desired shape is reached [20,34]. There are several technologies for this process, for example the FDM (fused deposition modeling) process, the most common being associated with the extrusion of thermoplastics. However, there are research and investigation opportunities for other processes, including the development of new materials and new technologies, within this same process [35].

Awasthi et al. [36] addressed the challenges faced by the FDM technique, given the wide variety of new materials emerging, emphasizing thermoplastic elastomers (TPE) that are compatible with the process. However, the process is limited due to the difficulty of printing, as well as the generation of defects in the produced parts. In the work by Jin et al. [37], a mathematical model was made and analyzed to verify the surface of the product manufactured through the FDM technique, indicating that the precision of the upper surface is determined through the ratio between the flow of the molten material and the feed rate of the nozzle, whilst the lateral surface is related to the thickness of the process layer and the bedding angle. Thus, it is suggested that the aforementioned ratio be properly used and the thickness of the layer is kept as thin as possible.

Thus, in that work, the process parameters were split into two groups: the first being the pre-processing parameters and the second the manufacturing parameters. Seven different proportions between Q (filament flowrate) and f (feed rate) were selected. It was observed that with increasing the ratio between Q and f, the surface quality became more consistent, but worsened when the ratio reached a value greater than the reference value of 0.585.

Yap et al. [38] used the additive manufacturing technique of material jetting to create a methodology that analyzes the capability of this process regarding the ideal quality and the dimensional accuracy of the manufactured part, utilizing three specific benchmarks. On the other hand, Tyagi et al. [39] made a thorough analysis of the material jetting technique, investigating the fundamentals of the process, as well as the characteristics and properties of the produced parts, especially the influences on mechanical properties, emphasizing that the orientation used is crucial to obtain better mechanical properties.

In relation to the powder bed fusion (PBF) technique, a heat source is used to melt the material in powder form, and thus giving rise to three-dimensional objects layer by layer, and with highly complex geometries [40–42]. This technology is often used to produce parts for the aerospace industry [43]. The selective laser sintering (SLS) process is the main technique behind this process [20], and the most suitable when it is desired to produce on a large scale [44]. Based on this, Nar et al. [45] characterized the surface topography

of the LS PA12 specimens. Compared to the injection molding technique, SLS produces surfaces with high roughness, due to the nature of the powder used, and this can affect their performance in its various features.

Another technique of additive manufacturing is called sheet laminating, and as the name implies, metal sheets are joined to form the product. UAM (ultrasonic additive manufacturing), a hybrid technique, is included in this class [46]. Figure 3 exemplifies this process [47].

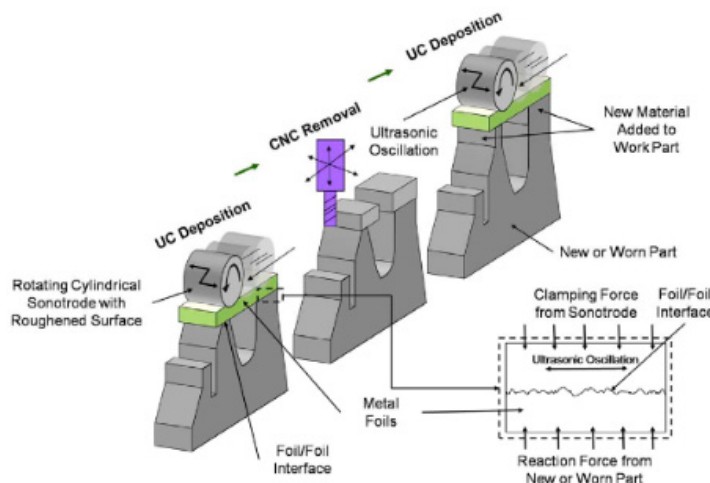

**Figure 3.** Representative image of the UAM process. Reproduced from [47]. Creative Commons CC BY license: https://creativecommons.org/licenses/by-nc-nd/4.0/; accessed on 21 October 2022.

Hehr et al. [48], in their work, thoroughly analyzed the entire history of the UAM process, in the case of the technological developments achieved and the areas in which it is most used. Knowing that the technique uses ultrasonic energy with the function of bonding the metal layers, the authors explained in a chronological way all the advances of the technique, as well as the characterization methods and how they influence the mechanical properties obtained.

Still in relation to additive manufacturing, another technique used is called vat polymerization. In this technique, UV or laser light is used in the photopolymerization of a liquid photopolymer, which solidifies, layer by layer, forming the product in 3D [49,50]. Revilla-León et al. [51], analyzed the influence of the design (solid, alveolar, and hollow) of the mold base with two different thicknesses (1 and 2 mm), in order to evaluate the accuracy of the obtained mold. For this, a digital mold of each design was made, and the honeycomb and hollow designs were further subdivided into the two thicknesses mentioned above. Additionally, a comparison of the mold obtained via printing with the digital mold was made. It was observed that the vat polymerization process can provide a good complement to the extrusion material process, being used for the manufacture of surgical instruments and in the dental field as well [52].

Based on this, it is known that nowadays, additive manufacturing is an industrial area with a last-generation of metal or composite printing machinery, used in an under-development sector that often uses cheap plastic filament printers to reduce production costs [14]. However, the continuous growth and studies around additive manufacturing demonstrate that it may have a significant place in the future of the industry, given the possibility of manufacturing lightweight and resistant materials and products, thus being able to be widely used in the aerospace as well as in the automotive industries, which requires high dimensional accuracy. In addition, in medicine, there has been a great advance when using this technology due to the possibility of producing bone prototypes and tailor-made protheses. Furthermore, in 2012, at least four additional significant technologies were observed in the additive manufacturing sector, which confirms its great growth and development [53].

As examples of recent advances, there is, according to the work of Khondoker et al. [54], an extruder that can carry out the deposition of mechanically interlocked extrudates composed of two immiscible polymers, which prints two filaments through a single nozzle, and with this, a reduction in adhesion failures between the filaments is observed. Furthermore, Cresswell-Boyes et al. [55] developed three-dimensional (3D) precise artificial teeth with two different materials (polylactic acid and thermoplastic elastomer) through tomography files generated from scans obtained by X-ray microtomography (XMT), using several open-source programs. The next approach will be to replicate the properties of natural teeth. In turn, Park et al. [56] demonstrated for the first time the multifunctional integration of various semiconductor devices produced through additive manufacturing, with the production of high-performance polymeric photodetectors and integrated multifunctional devices. Figure 4 illustrates this work.

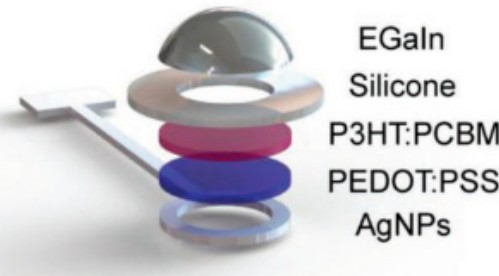

**Figure 4.** Structure of photodetector. Reproduced from [56], 2018, Wiley.

Therefore, additive manufacturing, according to Tofail et al. [57], is at a crossroads between a promising but unproven process for producing functional parts. Therefore, as observed by Pragana et al. [58], new applications have been developed in the field of additive manufacturing to overcoming the limitations of the several developed processes. Based on the factors that permeate this technology in development and growth, a brief diagnosis of this context was carried out through a SWOT analysis/diagram, as shown in Figure 5.

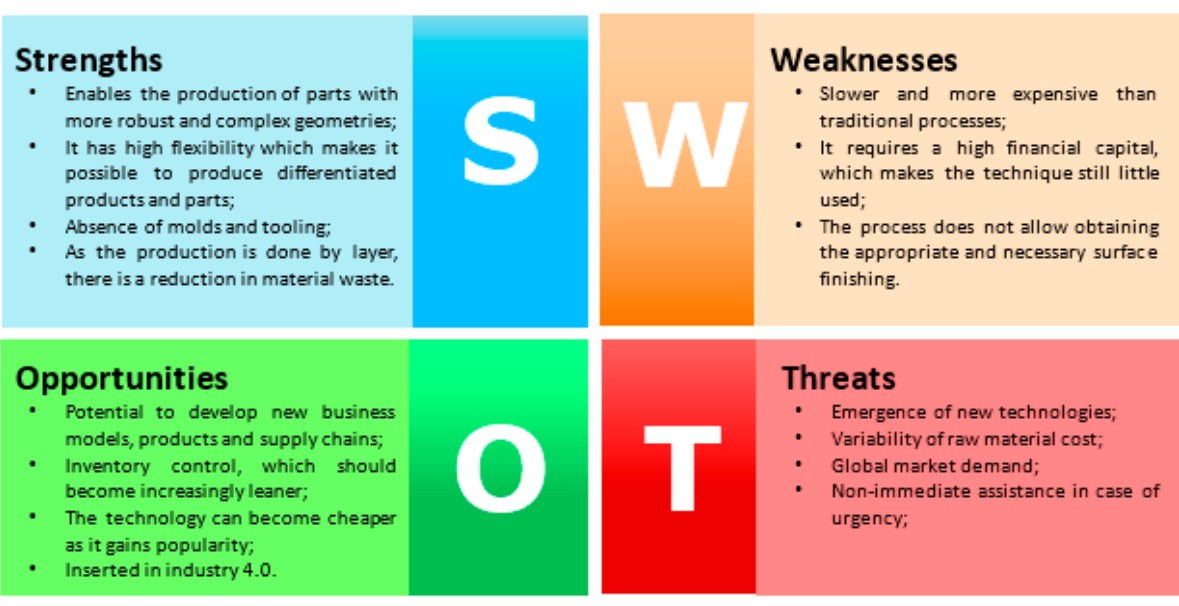

**Figure 5.** SWOT analysis regarding additive manufacturing.

## 3. CNC Machining

With the increase of industrial competition and energetic costs, it is very important for companies to reduce the production cycle and product prices, while ensuring the good quality of the parts. This is one of the reasons why companies should use CNC machines, since they can achieve high accuracy and short times in production/processing [59]. A CNC machine can be defined as a manufacturing process that is controlled through a computer with CAD and CAM systems, with the entire operation being carried out on the machine [60,61]. Figure 6 illustrates this process.

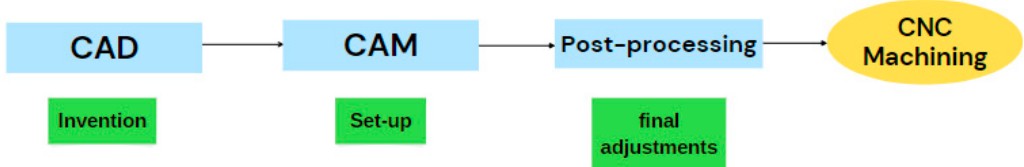

**Figure 6.** Representation of the CAM-CAD process on a CNC machine.

As in any process, some advantages and limitations can be observed. The advantages are good surface quality and dimensional tolerance, being ideal for high-end applications, in addition to the ability to process complex profiles, which would be hard to produce using conventional methods. In terms of limitations, there are the high cost of acquisition of the equipment and the need of specialized operators [62–65].

Gray et al. [66] has compared 3-axis machining and 5-axis machining, concluding that a 3-axis machine provided with an additional rotary/tilt table would improve the surface finish. On the other hand, Sheen et al. [67] developed a method to reliably determine the characteristics of the part to be machined between the upper and lower profiles. These characteristics have been used to program the subsequent manufacturing sequence that can be organized automatically with several generated cutting tool paths.

In the work by Senatore et al. [68], the recent search for cutting toolpaths on freeform surfaces was performed. The authors aimed at increasing productivity through a parallel plane milling strategy, determining an indicator to assist in choosing the best cutter suitable for end milling free-form surfaces. Furthermore, in the work by D'Souza et al. [69], an algorithm was designed to find the sequence of tools with the lowest cost in a 3-axis milling machine for machining free pockets, and further, the method can be adjusted to suit the set of tools available for milling.

4-axis machining has one more rotation for the X axis, called the A axis, which can cause the part to rotate, thus being able to be machined on 5 sides, and being more economical than 3-axis machining [70]. Axinte et al. [71] carried out a theoretical and experimental analysis regarding the functional feasibility of a 4-axis MMT (miniature multi-axis machine tools), reaching the conclusion that the micro-equipment can process small parts with satisfactory precision. Ding et al. [72] based his work on the contour EDM machining of free-form surfaces, thus developing a method capable of imposing tool paths for rough milling 4-axis contour EDM with a cylindrical electrode.

Using a 3-axis CNC machining center and incorporating a 4th axis via a horizontally oscillating rotary table, Ligten et al. [73] focused their work on producing aspherical surfaces, which were produced on this machining center in less than 15% of the time needed in the optical industries. Furthermore, it was observed in the work by Tang et al. [74] that an obstacle occurs when trying to maximize the number of machined surfaces in a 4-axis numerical control (NC) machine, and with that, the authors proposed a time algorithm $O((E+Iwb)2N)$, in order to minimize the number of setups needed to machine a part on a 4-axis NC machine.

Furthermore, post-processing technology is the crucial point for CNC automatic programming technology, as it is directly influenced by the processing quality and the production efficiency. Thus, Magambo et al. [75] created a post-processor for CNC systems,

indicating that their use generates better production efficiency and reliability. On the other hand, Baghi et al. [76] compared the effects of post-processing machining and annealing at 850 °C on titanium (Ti64) parts manufactured via selective laser melting (SLM). It is worth noting that each post-processing method acts in a certain way in relation to orientation, whether vertical or horizontal, for example. The fish scale defect was verified in samples that were only machined, being considered a failure of vertical samples, or even anisotropy of elongation between the vertical and horizontal samples that reduced by 125% on machined samples to 36% on annealed samples. Still in relation to post-processing, Chen et al. [77] analyzed hypoid gears because their manufacturing process results in large errors, since a simplified model of blades of cutting systems is used, and with this, a generic approach to programming and CNC post-processing was proposed that can be applied directly in the manufacture of these gears with great efficiency and superior quality.

*Machining with 5 or More Axis*

5-axis machining, in addition to the X, Y, and Z axes, has two rotary axes, which greatly improves the machining efficiency and accuracy. This process is commonly used to machine blades, rotors, dies, molds and propellers, among others [78–80]. As a result, it is considered the most important piece of equipment related to cutting in the industries, which allows the production of final components with much more complex shapes, and which would be impossible to obtain in a 3- or 4-axis machining [81]. Figure 7 exemplifies a 5-axis tool with table tilt A-C and X, Y, and Z axis, showing the flexibility presented by these systems.

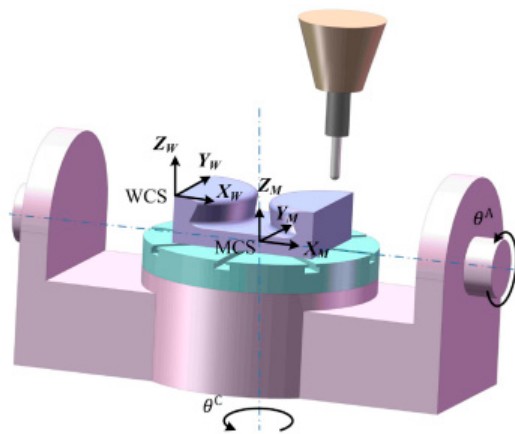

**Figure 7.** 5-axis tool with table tilt A-C and X, Y, and Z axis. Adapted from [78], 2021, Elsevier.

In the study by My et al. [82], a mathematical model was proposed that would be able to analyze and compare the kinematic performance of six configurations of 5-axis machines. Xu et al. [83] emphasized the great need to avoid changing the tool orientation in 5-axis machining. Thus, a smoothing method oriented to kinematic performance was proposed in the work.

Using a recent methodology called double-flank milling, Bizzarri et al. [84] investigated the fabrication of screw rotors, since the methodology was applied in 5-axis flank machining, and it was observed that for symmetrical profiles, double-flank milling is possible, and it works as a designed tool. Using this same methodology, Bo et al. [85] demonstrated a customized tool for machining narrow and curved regions with high precision, and the created algorithm was validated using commercial software.

Prabha et al. [86] used the Unigraphics NX6 CAD/CAM software, which is integrated into 5-axis CNC machines, to machine steam turbine blades, and then make the measurement through 3D coordinates, observing the great efficiency of the 5-axis machine in relation to dimensional accuracy. On the other hand, Huang et al. [87] emphasized the geometric errors occurring in a 5-axis machine, despite the precision of the machine,

and defined two models: "Rotary axis component displacement" and "Rotary axis line displacement", which, analyzed through the machining of five axes, has presented discrepant results. Therefore, the accuracy of 5-axis machining is the determining factor for success in processing and manufacturing parts and products [88]. Furthermore, it has a great advantage in terms of its flexibility and efficiency [89].

In 6-axis machining, one more rotation is added. Indeed, adding one more axis expands the variety of movements and transitions, reducing cutting time. Therefore, the 6-axle machine is used for machining very complex geometries, such as turbines or engine blocks [90]. Moriya et al. [91], through 6-axis control non-rotating cutting tools, processed curved V-shaped microgrooves on a curved surface. Due to the high difficulty of machining sharp corners using 3- to 5-axis machining, Japitana et al. [92] created a method for this process through 6-axis machining with ultrasonic vibration cutting, thus being able to create sharp corners on a protruding surface. Krimpenis and Noeas [93] also referred to the advantages of the machining process associated with additive manufacturing in the microfabrication, providing an insight into how these processes can be integrated.

Yuanfei et al. [94] designed a 6-axis CNC system through open architecture based on an industrial personal computer (IPC) and digital motion controller (DMC), thus analyzing the CNC design as well as the developed software. In turn, Carpiuc-Prisacari et al. [95] used a 6-axis machine to perform an analysis on the propagation of mixed-mode cracks; for this it was necessary to initialize a crack, and its propagation through rotation, traction and shear was obtained by the 6-axis machine. Based on robotic machining with six degrees of freedom, Huynh et al. [96] modeled several flexible industrial robots with six axes to be used with milling operations, with torsion springs and dampers positioned on each axis to assist in flexibility, since the lack of joint stiffness is a possible limitation. Milling operations on aluminum and steel corners were simulated, the latter being unsatisfactory because the results did not correspond to the experimental results.

5-axis CNC machining becomes even more critical when machining difficult-to-machine materials, where the vibrations generated and the tool moving simultaneously in different axes can cause stiffness problems, causing imperfections in the machined surface. Indeed, the machinability of the material to be machined strongly impacts the process results and machine behavior [97,98].

Thus, it is observed that the machining of parts has a very broad market, but with some limitations, such as waste of material, for example. Figure 8 illustrates the SWOT analysis for this process, emphasizing that the tool and CNC market represents a large share of the global market, and this aspect is of great importance for the process.

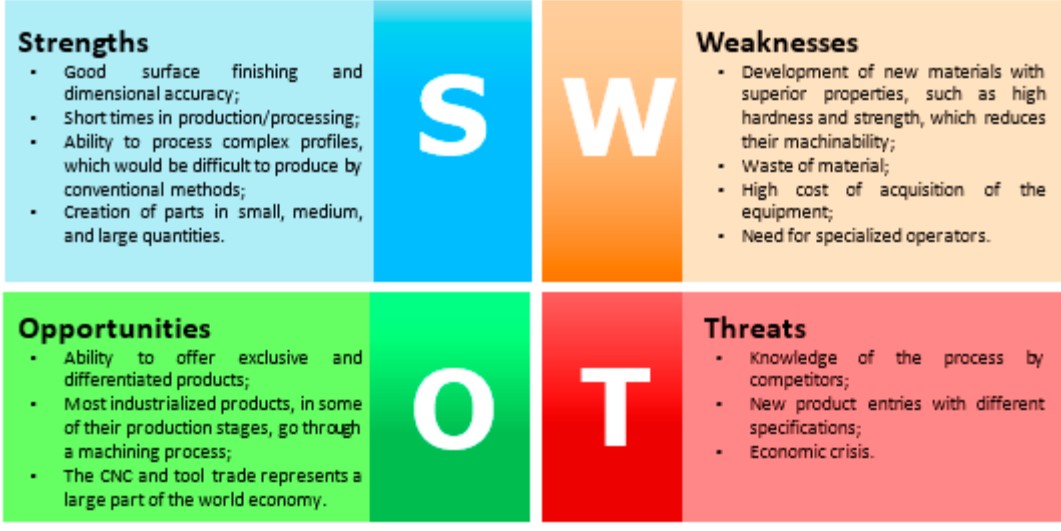

**Figure 8.** SWOT analysis regarding machining.

## 4. Hybrid Manufacturing Processes

Nowadays, due to the development of new materials with better mechanical properties, less specific weight and better performance, the machining industry is faced with an enormous challenge to machine those materials, and thus, new processes had to be coupled and applied to satisfy this need. This is the case in hybrid manufacturing processes [1,99]. The concept of hybrid manufacturing is basically defined as the manufacturing process that joins two or even more manufacturing methods in a single piece of equipment (Figure 9), which has a great effect on the overall performance of the process [1]. The typical example of hybrid manufacturing is the combination of additive manufacturing and machining, using powder bed fusion (PBF) or directed energy deposition (DED) as an additive process with 5-axis machining [100]. This process is especially important when it comes to materials that are difficult to machine. However, many studies still need to be carried out, especially when dealing with complex geometries in 5-axis machining [101].

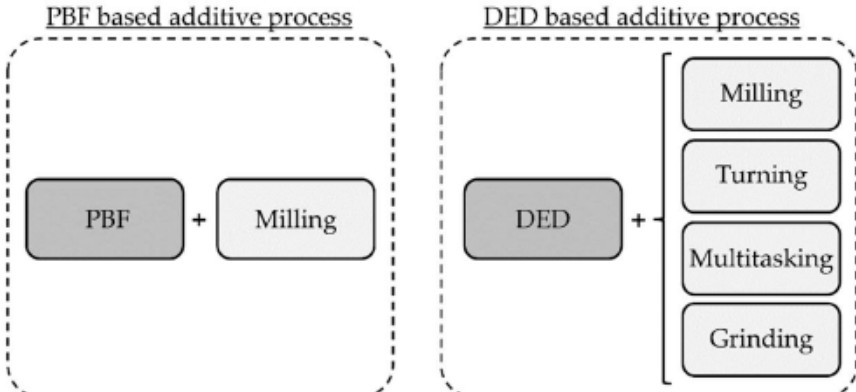

**Figure 9.** Different combinations of processes that can be performed using hybrid manufacturing, with powder bed fusion (PBF) or directed energy deposition (DED). Reproduced from [100]. MDPI Open Access Information and Policy: https://www.mdpi.com/openaccess; accessed on 21 October 2022.

Soshi et al. [102], in their work, manufactured a prototype of a mold normally produced by injection through a hybrid process involving DED technology as an additive process, followed by milling for surface finish and subtractive process. What could be observed in relation to the cooling performance of the mold is the uniformity of this cooling in relation to the traditional process and more stable temperature throughout the cycle, thus improving the cooling performance. In turn, Chen et al. [103] proposed an algorithm capable of calculating the machinability of the material and the product, determining as well the minimum number of alternations between additive and subtractive processes to form a highly complex part and guarantee an ideal cutting tool path in the subtractive operation. The algorithm was called Top-Down_Sequential_Maximization; however, only 3-axis machining was used by default.

Flynn et al. [104] addressed the issue of additive and subtractive hybrid machines through a careful review of the literature, considering the DED method in conjunction with CNC machining, with this being proposed in a future vision that forms a closed circuit. Li et al. [105] developed a 6-axis hybrid additive–subtractive manufacturing process, using a robotic arm with six degrees of freedom, together with a platform for manufacturing parts, and observed an improvement in the surface quality of the part, reduction of material waste and production time. In addition, this enables the need for a support, given the flexibility found in the six axes. Figure 10 illustrates the configuration used.

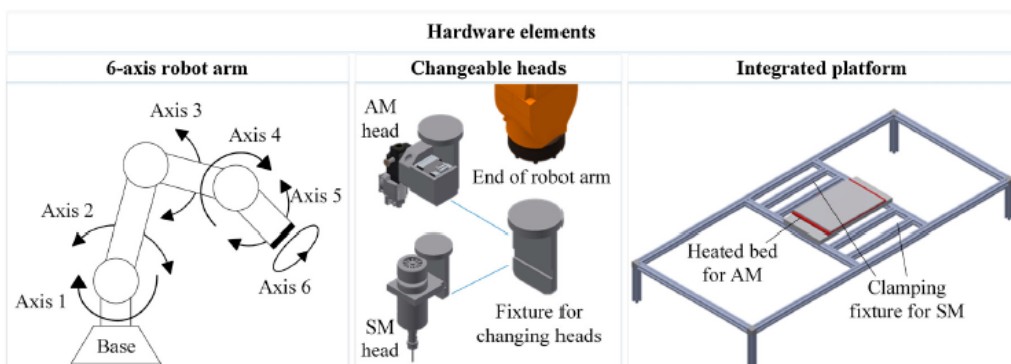

**Figure 10.** Configuration used in the study for the hybrid additive–subtractive manufacturing process. Reproduced from [105], 2018, Elsevier.

Following this lead, Yamazaki et al. [10] developed a hybrid manufacture system integrating the additive method LMD (laser metal deposition), with turning and milling tools, through the Mazak Hybrid multi-tasking machine and exposed application examples for this process, such as the oil industry. With this, it was possible to notice a huge advantage over the standard manufacturing process.

Chen et al. [106] implemented planning algorithms capable of combining additive technology with subtractive technology. In this case, in the first stage a subtractive technology was used to create the beginning of the product's geometry, followed by additive manufacturing, and ending with a machining process for surface finishing. The additive process used was the DED method and a 5-axis machine, more specifically the 3+2-axis machine. Newman et al. [107] worked with a structure called iAtractive, based on which a system called Re-Plan was created for process planning, and with it analyzed the capacity for the integration between additive and subtractive processes through case studies. With Re-Plan, it was possible to perceive that according to the geometry and complexity of the part, the material is added or removed, and even the material of an existing part can be reused to form a product with a new identity.

From a perspective considering the energy expenditure of the process and environmental damage, Yang et al. [108] analyzed the energy consumption performance when using a 6-axis robotic arm as a subtractive process, after manufacturing via additive manufacturing using the fused deposition modeling process. Several case studies and scenarios were used, and the result obtained was quite complex. Regarding future work, it is intended to carry out a mathematical and numerical analysis between the configuration of the robot arm and the energy consumption.

Liou et al. [109] combined the laser deposition process and a 5-axis CNC milling system through process planning and visualization, aiming at integrating all the processes. Additionally, Grzesik [110] highlighted additive–subtractive hybrid manufacturing through the techniques of LMD (laser metal deposition) and multi-axis CNC machining. Furthermore, Ren et al. [111] used the hybrid machining technique with DED and 5-axis machining to repair dies, targeting the corroded and worn surface. In addition, several other authors have based their studies on hybrid manufacturing, and Table 1 summarizes some pertinent works.

**Table 1.** Summary of studies about hybrid manufacturing, coupling addictive processes and machining.

| Author/Year | Technique | Material | Description |
|---|---|---|---|
| Cococcetta et al. (2021) [112] | 3D printed + Dry, MQL and Cryogenic machining | CFRP | Analysis of printing and machining parameters and cooling/lubrication conditions in the production of printed CFRP thermoplastic compounds, from the comparison of three post-processing methods: dry, minimum quantity lubrication (MQL) and cryogenic machining. |
| Tapoglou et al. (2020) [113] | DED + Milling | 316L Steel | Production of 316L stainless steel parts using the DED technique with subsequent machining, from defining the best parameters for material deposition, to analyzing the machinability of the deposited material. |
| Hällgren et al. (2016) [114] | PBF + HSM | Aluminium | Analysis of the cost of serial production of parts if print speed increases, if machine cost or part mass is reduced, using GMP techniques such as additive manufacturing and high-speed machining (HSM). |
| Kaynak et al. (2018) [115] | SLM + FM/VSF/DF | 316L Steel | In order to improve the surface quality of the parts manufactured via selective laser melting (SLM), three post-processing techniques were performed: finishing machining (FM), vibrating machining surface finishing operations (VSF) and drag finishing (DF). The surface was analyzed to meet quality requirements, and it was verified that the DF method resulted in a less rough and more consistent surface finish. |
| Bai et al. (2020) [116] | SLM + Milling | 6511 Steel | Analysis of the production of 6511 martensitic stainless steel parts through selective laser melting (SLM) and end milling, optimizing the process parameters, and taking into account the residual stresses arising from the phase transformation of the martensitic steel. |
| Kaynak et al. (2018) [117] | SLM + Machining | Inconel 718 | Study of surface finish through machining, including surface roughness, microhardness and XRD analysis of Inconel 718 alloy, produced via SLM, verifying that the roughness had a decrease of 90% after post-processing with machining. |
| Salonitis et al. (2015) [118] | Laser Cladding + HSM | Steel | Quality verification of a steel tube manufactured via laser cladding after the high speed machining process, which reduced residual stresses and distortion from the additive manufacturing process. |
| Pal et al. (2016) [119] | DMLS + Machining | SS PH1 | Characterization of the mechanical properties of the PH1 stainless steel product produced via direct metal laser sintering (DMLS) in conjunction with machining process parameters and post-processing. |
| Careri et al. (2021) [120] | DED + Machining | Inconel 718 | Evaluation of surface finish, microstructure, microhardness and residual stresses from the manufacture of parts in Inconel 718 by DED, followed by two possible post-processing: machining + heat treatment or heat treatment + machining, verifying that the best production strategy was AD + M + DA. |
| Heigel et al. (2018) [121] | PBF + Machining | Stainless steel | Production of stainless steel cylinders through the laser powder bed fusion process, followed by machining, and analysis of residual stresses from the processes. |
| Speidel et al. (2021) [122] | PBF + EJM | Ti-6Al-4V | Due to the poor surface quality due to unmelted powder from the powder bed fusion process, this study analyzed the application of electrochemical jet machining in order to produce a good surface finish and increase the functionality of the part. |
| W. Grzesik [123] | General | General | This work provides a recent and brief overview of hybrid machining. |

As noted, hybrid manufacturing brings numerous advantages and benefits. However, some challenges are still encountered, as depicted in Figure 11 [100].

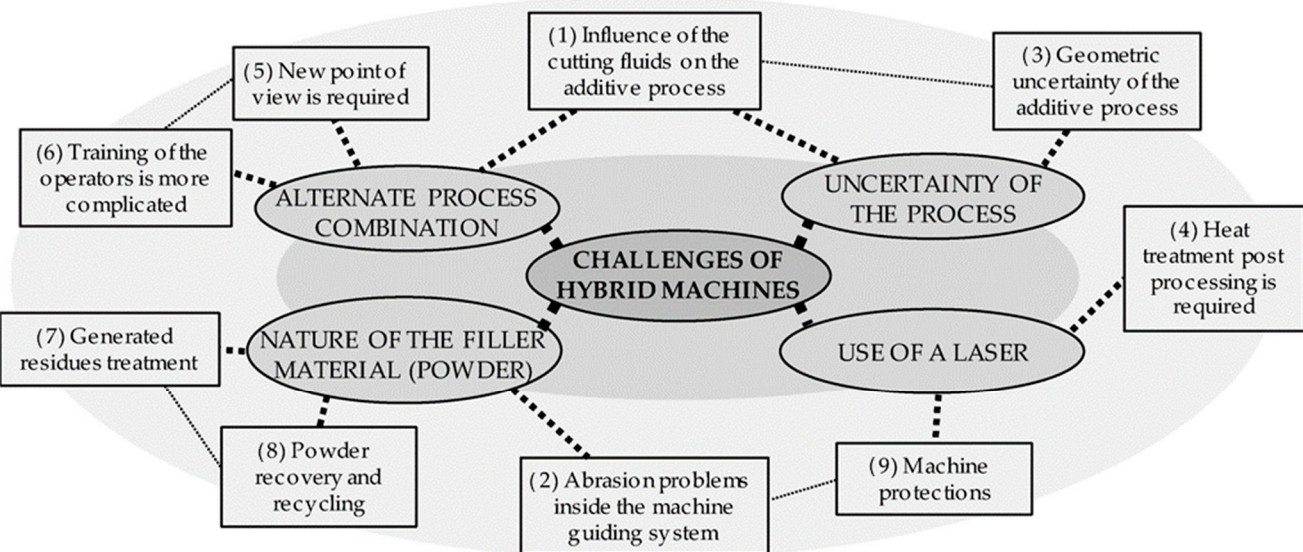

**Figure 11.** Schematic figure illustrating the challenges of hybrid machining. Reproduced from [100]. MDPI Open Access Information and Policy: https://www.mdpi.com/openaccess; accessed on 21 October 2022.

As a challenge to be faced, the recycling of metallic powder can be considered, since it can be harmful to human health, especially nickel or cobalt compounds, and must be removed from the machine after the addition operations are carried out, so that they do not affect machine components in order to avoid damage and breakdown [7]. In addition, the protection of the machine, safety and reliability of the process must be considered, since due to the heat generated during the additive process, it can result in the melting of some areas, which must have adequate protection [98]. Additionally, when talking about the challenges faced when implementing hybrid production, it is important to emphasize the training of qualified technicians so that they are able to understand and apply both techniques, and the correct disposal of waste, with a focus on handling the waste, dust and recycling liquid waste, such as lubricating oils and cutting fluids. Another problem faced is precisely in the use of these cutting fluids, since laser technology was developed regarding a fluid-free environment, and on the other hand, in machining this is widely used and practically essential or mandatory for some materials to be machined [7,98].

As hybrid manufacturing is a process that is still under development and expansion, many challenges are encountered and need to be resolved, both in relation to scientific and research parameters, as well as the technical parameters for its implementation on an industrial scale [124]. In this sense, some questions must be answered, such as, for example, what would be the best processing sequence, how the microstructure and properties of the part would behave after the cycles, or even how the software used would be able to detect collisions after adding more material [125].

However, according to Dilberoglu et al. [126], it is expected that in the future the use of hybrid production systems will be increased with additional improvements, as the implementation of this technology has attracted the attention of researchers in modern industries. Thus, based on the current scenario, and with the analysis carried out in this work, the following SWOT matrix for hybrid manufacturing resulted (Figure 12).

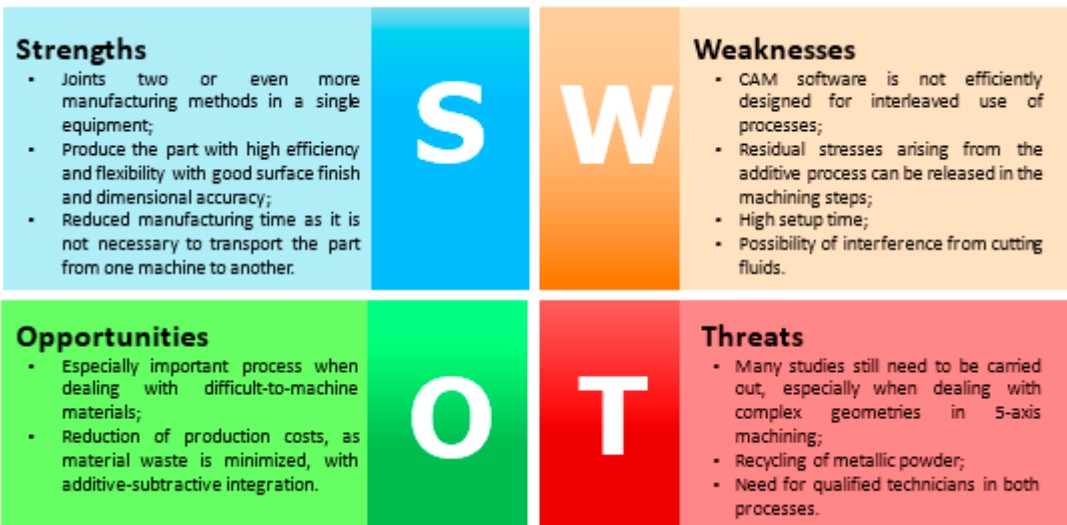

**Figure 12.** SWOT analysis corresponding to hybrid manufacturing.

Through this review work, it was possible to verify that there is a lot of work to be done in the field of hybrid manufacturing, which incorporates some developments carried out in each of the integrated processes (additive manufacturing and machining). This was evident through this study, which seeks to provide a structured, broad and integrated view of the contributions that have been made lately by several researchers in each of the areas. Additionally, it was possible to see how other researchers have taken advantage of these developments, and taking the knowledge acquired in each one of the considered processes, to use it in the integration of the two processes.

There is still a lot of work to be developed in the area of hybrid production. The interaction of the additive and subtractive processes does not have to be sequential and carried out in this order, although the additive process needs to be the first, but it can be reused later if there are problems with accessing the tools in certain areas. Indeed, the parts tend to have an increasingly complex shape, which also requires increasingly refined techniques for their manufacture. CAM software has not yet been developed efficiently for an interleaved use of processes, and this interleaved use may be absolutely necessary to realize different geometries. This is certainly a subject that will be of interest to researchers in the near future. Another issue to be aware of is the level of internal stresses left on the parts after machining. Although the volume of chips to be removed is generally small, finishing operations generally raise the temperature, which may influence the mechanical properties of the material being worked. Thus, there may be overlapping stresses in parts of the parts, which can lead to deformations that may interfere with the functionality and expected lifetime of the machined part. Thus, the induction of stresses and their level are also subjects that will certainly be investigated in the near future.

### 5. Conclusions

Additive manufacturing is undoubtedly an effective and emerging method for the manufacture of complex and difficult parts to be produced by other methods; however, a major limitation of the processes involved is the poor surface quality and dimensional accuracy, requiring a subtractive method to achieve the required surface quality and shape tolerance.

Thus, hybrid manufacturing has gained great prominence, bringing many benefits to modern industry. It guaranteed, through the integration between additive and subtractive processes, the production of parts that would previously be difficult to be produced with only one of the individual techniques, either due to the high hardness of the material, or because of the high complexity and dimensional tolerance.

One of the great advantages of this process is the high flexibility and versatility, as well as the reduction of waste, a problem highly faced by conventional machining. In addition, the possibility of using only one machine during the entire process eliminates transport times and process inventories. As a limitation, we can highlight the residual stresses present in the additive manufacturing process, which can result in distortions in the part, requiring further heat treatment.

In addition, in terms of challenges for this technology to be implemented, we can mention the recycling process of metallic powder, reliability and safety in the process, and training. Indeed, the proper disposal of waste, as well as the use of cutting fluids in machining remains an environmental issue, since the additive manufacturing technology-using laser was developed applying environmentally friendly means, free of these fluids.

Based on this, this work has shown through SWOT analysis and a broad bibliographic review the studies and developments, as well as the great growth of this process, thus contributing to the study and analysis of the development of new design of parts, which previously would be impractical, and today are already a promising reality. Moreover, the combination of both technologies helps in overcoming the traditional issues related to the surface finishing of additively manufactured parts, providing the surface quality usually required in many applications. In this way, it is expected that in the future there will be a greater integration of hybrid machines in production lines, so that they can produce parts that are closer to the final product shape without the need for post-processing.

In the future, it is expected that the efforts to integrate the two technologies will evolve positively in the sense of increasing the rigidity of the equipment, the versatility of the transition from one technology to the other, and the complementarity in terms of work in both technologies. In fact, it is possible to take advantage of the knowledge acquired in terms of surface quality left by the additive manufacturing and try to minimize surface defects, thus minimizing the machining work to be carried out later. Furthermore, 5-axis machining makes it possible to take better advantage of topological optimization, making more complex shapes in a single fixture, thus increasing the accuracy obtained. The training needs are evident, as the requirements at this level start with the design of the parts, and end with obtaining the final parts in the equipment. Thus, there will have to be a greater integration of knowledge between those who develop the products and those who make them possible through the programming and operation of hybrid equipment. The improvement of these factors and a natural reduction of investment costs in the near future, will be decisive for the rapid growth of this technology in the market.

**Author Contributions:** Conceptualization, N.P.V.S. and F.J.G.S.; methodology, N.P.V.S.; validation, F.J.G.S., F.F. and V.F.C.S.; writing—original draft and designed, N.P.V.S.; writing—review and editing, F.J.G.S., F.F. and V.F.C.S.; Supervision, F.J.G.S.; Funding acquisition, F.J.G.S. and F.F. All authors have read and agreed to the published version of the manuscript.

**Funding:** The present work was done and funded under the scope of the projects ON-SURF (ANI | P2020 | POCI-01-0247-FEDER-024521 and MCTool21 "Manufacturing of cutting tools for the 21st century: from nano-scale material design to numerical process simulation" (ref.: "POCI-01-0247-FEDER-045940") co-funded by Portugal 2020 and FEDER, through COMPETE 2020-Operational Programme for Competitiveness and Internationalisation. This work is also sponsored by FEDER National funds FCT under the project CEMMPRE ref. "UIDB/00285/2020". F.J.G. Silva also thanks INEGI-Instituto de Ciência e Inovação em Engenharia Mecânica e Engenharia Indústria due to its support.

**Conflicts of Interest:** The authors declare no conflict of interest. The funders had no role in the design of the study; in the collection, analyses, or interpretation of data; in the writing of the manuscript, or in the decision to publish the results.

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
