# Peer review of "Hybrid Manufacturing Processes Used in the Production of Complex Parts: A Comprehensive Review"

_metals, doi:10.3390/met12111874_

Round 1

Reviewer 1 Report

The paper try to address very interesting and actual topic connected with integration of additive and subtractive processes. Unfortunately, the presented contents can not be considered as review. The majority of the manuscripts consist of separate discussion of main additive and CNC machining processes (by the way very basic). The paragraph connected with integration of these process is limited to 2-3 pages and table (which can not be considered as review). In my opinion Authors did not provide a complete and synthetic analysis covering topic of the work, the discussion of the significance of carried out research for further development in this area is limited. Therefore, the submitted manuscript does not meet the standards of good review paper and should be rejected.

Reviewer 2 Report

This article provides very interesting information and it is suitable for publication in “Metals” after revision according to the comment below:

1)It is not necessary to provide general information about additive manufacturing, UAM process or CNC machining, I suggest removing these parts

2)You should provide more information about the current state of the art in this issue, e.g. Comprehensive analysis and study of the machinability of a high strength aluminum alloy (EN AW-AlZn5.5MgCu) in the high-feed milling;  Prediction of Cutting Material Durability by T = f(vc) Dependence for Turning Processes; or  Hybrid additive and subtractive manufacturing processes and systems: a review, etc.

3)You should provide general state of the art with using some special methodology e.g. PRISMA, or QUOROM

4)Table 1 is very long and chaotic; you should try to replace it with a figure

5)Conclusion – Please provide a contribution of your findings to practice and science

Reviewer 3 Report

The paper entitled " Integration of additive and subtractive processes for production of complex parts: a review " presents an extensive literature review of the additive-subtractive integration in order to fabricate complex parts. From my point of view, the article is of great interest but more analysis could be done to improve overall quality:

·        A graphical abstract would add interest to catch the eye

·        It would be interesting to introduce beforehand that the WAAM technology belongs to the direct energy deposition classification of the standard.

·        I think it would be interesting to include with the machining of WAAM material and the Post- Processing:

o    Process Parameters Optimization of Thin-Wall Machining for Wire Arc Additive Manufactured Parts. Applied Sciences. 2020; 10(21):7575. https://doi.org/10.3390/app1021757

o    Wire arc additive manufacturing Ti6Al4V aeronautical parts using plasma arc welding: Analysis of heat-treatment processes in different atmospheres. Journal of Materials Research and Technology, 9(6), 15454–15466. https://doi.org/10.1016/j.jmrt.2020.11.012

o    Analysis of the machining process of titanium Ti6Al-4V parts manufactured by wire arc additive manufacturing (WAAM). Materials, 13(3). https://doi.org/10.3390/ma13030766

·        One topic with recent contributions that would be interesting to discuss is that of topological optimization. In the case of WAAM is not the most suitable, although, there are several paper dealing with this topic:

o   https://doi.org/10.1089/3dp.2021.0008

o   https://doi.org/10.1016/j.jcsr.2021.106887

o   https://doi.org/10.1016/j.addma.2019.06.010

o   https://doi.org/10.1007/s11665-022-06702-x

·        please avoid the use of 1st person sentences like: “we can see the advantages offered by hybrid manufacturing, compared to 458 the addition and subtraction techniques processed separately…”

·        Please add some future lines that could add a vision on next steps

These comments are intended to contribute to the improvement of the quality of the paper presented.

Reviewer 4 Report

The authors presented an article «Integration of additive and subtractive processes for production of complex parts: a review». However, there are several points in the article that require further explanation.

Comment 1:

The abstract needs to be improved.

Demonstrate in the abstract novelty, practical significance. What is the difference between your article and previously published reviews on this topic? What has been done and reviewed here for the first time? Why will readers be interested in this particular review? It is important to show the relevance more clearly. It will be useful to list which AM methods are discussed in the article. What machining methods are discussed in the article.

Comment 2:

The introduction needs to be improved.

Relevance, comparison with previously published review articles, what is the content of the article?

It will be useful to cite review articles corresponding to the topic of the article in order to show what they have not previously done by previous scientists on the topic of the article.

Without this, the introduction looks weak.

After analyzing the literature, show before formulating the goal of the "blank" spots. Which has not been previously done by other researchers. You must show the importance of the research being undertaken. Show what will be the new research approach in this article. You need to show a hypothesis.

Show what is the scientific novelty of the article. What has been done for the first time?

Add a clear purpose to the article.

Briefly list what is done in each section.

Comment 3:

2. Additive manufacturing

3. CNC Machining

Are all figures original? If not needed appropriate citations and permissions. Refine this for figures throughout the article.

In AM methods it will be useful to consider:

Journal of Manufacturing Processes 2022, 80, 328-346. doi:10.1016/j.jmapro.2022.06.009

Materials 2021, 14, 3866. doi:10.3390/ma14143866

Comment 4:

Sections 4 and 5 will lack such a clear systemic analysis and approach.

But most importantly, at the end of each of subsections 4,5, it is necessary to add a critical analysis of the phenomena studied by the authors. It is not enough to enumerate the literature. And it is important to draw general conclusions. What challenges will this area face in the near future? What are the benefits of implementing this.

Comment 5:

Before Conclusions should be clearly labeled Challenges and Future trends.

Comment 6:

The conclusions seem rather sparse. And they do not allow the reader to fully show the perspectives of this topic. And they should be significantly reworked and strengthened.

The article is interesting, but needs to be improved. Authors should carefully study the comments and make improvements to the article step by step. After major changes can an article be considered for publication in the "Metals".

Round 2

Reviewer 2 Report

Dear authors,

thank you very much for your revision. In my opinion, this article is suitable for publication in present form.

Author Response

Thank you for your positive response

Reviewer 4 Report

The authors have improved the article. However, the most important and key points are still not disclosed by the authors.

1. The abstract still does not mention which additive manufacturing methods are considered in the article? SLM, SLS, etc. Same question for machining methods. But most importantly, there is still no clear understanding of what exactly is the novelty of the article, what new approaches are used, why it is important for the reader to read this review, etc.

2. Also, the authors must clearly justify the relevance and novelty of the proposed review. And to consider what exactly is the progress of this review over previously published articles, such as:

Application of Hybrid Manufacturing processes in microfabrication. 2022, 80, 328-346. doi:10.1016/j.jmapro.2022.06.009

Metal hybrid additive manufacturing: state-of-the-art. Progress in Additive Manufacturing 2022, 7(4), 737-749. doi:10.1007/s40964-022-00262-1

Hybrid metal additive manufacturing: A state–of–the-art review. Advances in Industrial and Manufacturing Engineering 2021,

2, 100032. doi:10.1016/j.aime.2021.100032

Current trends and research opportunities in hybrid additive manufacturing. International Journal of Advanced Manufacturing Technology 2021, 113(3-4), 623-648. doi:10.1007/s00170-021-06688-1

Hybrid additive manufacturing of steels and alloys

Open access. Manufacturing Review 2020, 7, 6. doi:10.1051/mfreview/2020005

Hybrid processes in additive manufacturing. Journal of Manufacturing Science and Engineering, Transactions of the ASME 2018, 140(6), 060801. doi:10.1115/1.4038644

Additive manufacturing: scientific and technological challenges, market uptake and opportunities. Materials Today 2018, 21(1), 22-37. doi:10.1016/j.mattod.2017.07.001

etc.

Such a selection and search for relevant review articles should have been done by the authors and not by the reviewer. And now the authors must briefly review each review article and show what progress they have made. Without this, this review does not have any novelty and "zest" that would distinguish it from previously published ones.

After analyzing the literature, show before formulating the goal of the "blank" spots. Which has not been previously done by other researchers. You must show the importance of the research being underway. Show what will be the new research approach in this article. You need to show a hypothesis. Show what is the scientific novelty of the article. What has been done for the first time? Add a clear purpose to the article. Briefly list what is done in each section.

Otherwise, the article cannot be recommended for publication. Publishing articles that repeat already known approaches does not make sense for an international journal. Therefore, authors should be very serious about giving clear answers and justifications to readers.

Author Response

Please see reply in attach

Round 3

Reviewer 4 Report

Now the article has been improved in accordance with the comments and can be accepted for publication.